# Transmission of Functional, Wild-Type Mitochondria and the Fittest mtDNA to the Next Generation: Bottleneck Phenomenon, Balbiani Body, and Mitophagy

**DOI:** 10.3390/genes11010104

**Published:** 2020-01-16

**Authors:** Waclaw Tworzydlo, Malgorzata Sekula, Szczepan M. Bilinski

**Affiliations:** Department of Developmental Biology and Invertebrate Morphology, Institute of Zoology and Biomedical Research, Faculty of Biology, Jagiellonian University in Krakow, Gronostajowa 9, 30-387 Krakow, Poland; w.tworzydlo@uj.edu.pl (W.T.); malgorzata.sekula@doctoral.uj.edu.pl (M.S.)

**Keywords:** Balbiani body, oogenesis, oocyte, mitochondria selection, mitophagy

## Abstract

The most important role of mitochondria is to supply cells with metabolic energy in the form of adenosine triphosphate (ATP). As synthesis of ATP molecules is accompanied by the generation of reactive oxygen species (ROS), mitochondrial DNA (mtDNA) is highly vulnerable to impairment and, consequently, accumulation of deleterious mutations. In most animals, mitochondria are transmitted to the next generation maternally, i.e., exclusively from female germline cells (oocytes and eggs). It has been suggested, in this context, that a specialized mechanism must operate in the developing oocytes enabling escape from the impairment and subsequent transmission of accurate (devoid of mutations) mtDNA from one generation to the next. Literature survey suggest that two distinct and irreplaceable pathways of mitochondria transmission may be operational in various animal lineages. In some taxa, the mitochondria are apparently selected: functional mitochondria with high inner membrane potential are transferred to the cells of the embryo, whereas those with low membrane potential (overloaded with mutations in mtDNA) are eliminated by mitophagy. In other species, the respiratory activity of germline mitochondria is suppressed and ROS production alleviated leading to the same final effect, i.e., transmission of undamaged mitochondria to offspring, via an entirely different route.

## 1. Introduction

Mitochondria are semi-autonomous organelles that contain their own genome, termed the mitochondrial DNA (mtDNA), and protein synthesis machinery. The most important role of mitochondria is to supply cells with metabolic energy (in the form of adenosine triphosphate (ATP) molecules) generated by oxidative phosphorylation. As the latter process, in addition to ATP, generates also reactive oxygen species (ROS), mtDNA is particularly vulnerable to mutations and lesions. Gradual accumulation of the mutations leads to deleterious effects including a loss of metabolic functions and decline in mitochondrial membrane potential. It is considered in this context that high potential of mitochondrial inner membrane indicates the integrity and functionality of mtDNA of a given mitochondrion. In somatic cells, the negative effects of ROS action are counteracted by two processes: mitochondrial fusion and fission that are collectively termed “mitochondrial dynamics” or “mitochondrial homeostasis”. Obliterating mitochondrial fission leads to the formation of extensive mitochondrial networks (hyperfused mitochondrial networks), whereas disruption of mitochondrial fusion results in maintaining mitochondria in the form of small bean-shaped, individual organelles. Thus, the actual morphology of mitochondria in a given cell depends on a balance between these opposed processes. It has been shown recently that both fusion and fission are implicated in maintenance and inheritance of mtDNA. Damaged mitochondria with low membrane potential might be rescued by fusion (with “healthy” mitochondria) or alternatively eliminated (via mitophagy) after fission, i.e., after separation from the mitochondrial network (see [1,2,3,4,5] for further details).

In most animal groups, mitochondria are inherited from one generation to the next maternally, in a non-Mendelian way, being exclusively transmitted from female germline cells, i.e., the oocytes and eggs. A notable exception to this general rule has been described in bivalve molluscs [6]. In this taxon, the female embryos inherit mitochondria from female germline cells (eggs), whereas the male ones from spermatozoa [7,8].

In organisms reproducing asexually, a phenomenon termed Müller’s ratchet leads to accumulation of irreversible deleterious mutations in subsequent generations. These mutations affect the viability of cells and then whole organisms. As in the vast majority of animal species, mitochondria are transmitted to the offspring only maternally (that is asexually), their mtDNA must escape from Müller’s ratchet or must be protected against the impairment. Although the mechanism of such escape/protection is far from being clear, it is widely accepted that it relies on the mtDNA bottleneck phenomenon. This phenomenon involves at least three, possibly cooperating, mechanisms: (1) random segregation (partition, sampling) of various mtDNA genotypes to early germline cells during proliferation (expansion) of the oogonial population, (2) elimination of individual mitochondria (or whole early germline cells) overloaded with mtDNA mutations, and (3) expansion of healthy mitochondria within the oocyte cytoplasm. Moreover, there is an ongoing vivid, and partly theoretical debate about which mitochondria are actually transmitted to the zygote: functionally highly active, i.e., those containing undamaged mtDNA molecules, or alternatively functionally silenced (to preserve the integrity of their mtDNA) (see [8] for further details). Recently, the first possibility (termed here, the high energy pathway or mitochondrial selection) seems to be more substantiated, whereas the second (the low energy pathway) is often brought into question. Here we discuss both pathways and present a new putative example of the second one.

## 2. Categories (Types) of Animal Ovaries

Animal ovaries fall into two morphologically and physiologically disparate categories (types): panoistic and meroistic. This dichotomous split must be taken under consideration while discussing pathways of mitochondria transmission from one generation to the next. Therefore, we will start this review with a very short description of main ovarian categories. For comprehensive and detailed description of all ovarian types and subtypes see [9,10,11].

Panoistic ovaries are characteristic for the majority of invertebrates, e.g., higher crustaceans, chelicerates, basal insect lineages (with some exceptions), and nearly all vertebrates. In this ovary category, all female germline cells develop into fertilizable gametes (egg cells). This implies that all the mitochondria of a given egg are recruited from a single oogonial cell. Meroistic ovaries have been found in rotifers, annelids, lower crustaceans, advanced (holometabolous) insects and certain lizards [9,12,13,14]. Here the oocytes develop within syncytial clones (clusters) of sibling germline cells. In each cluster, only one cell differentiates into the oocyte, while the others become supporting nurse cells or trophocytes. As oogenesis progresses, all the organelles and macromolecules present (and synthesized) in the cytoplasm of the nurse cells are transferred to the growing oocyte where they are indispensable for the formation of fully grown egg [11,12]. This transfer consists of two phases: slow and rapid that is also known as the nurse cell dumping [9]. Interestingly, it has been recently shown that murine ovaries are also meroistic and that oocytes of this mammalian species acquire organelles from sister cells [15,16]. It is obvious that in all animal taxa with meroistic ovaries (including mice), the mitochondria of a given egg cell originate from all interconnected sibling cells.

## 3. Selection of Highly Active (“Healthy”) Mitochondria and the Balbiani Body—The High Energy Pathway

As mtDNA suffers from high mutation rates and is inherited in a non-Mendelian way only from the mother, the mitochondria theoretically should accumulate deleterious mutations relatively quickly leading to the impairment and mutational meltdown of animal populations. Experimental data indicate that it is not the case, and that propagation of deleterious mitochondrial variants is restricted in both vertebrates [17,18] and invertebrates [19]. Consequently, functional mitochondria and the fittest mtDNA are preferentially transmitted to the next generation. Several lines of evidence suggest that selection and proliferation (replication) of the healthy mitochondria is associated with a female specific organelle assemblage, termed the Balbiani body or mitochondrial cloud [19,20,21].

The Balbiani body (Bb) is a transient complex of various organelles positioned at one side of the oocyte nucleus and often closely associated with its envelope (Figure 1A–C) (see [22,23] for further details). The structure as well as composition of the Bb may differ even in related animal lineages and are rather dynamic in subsequent stages of Bb morphogenesis, i.e., its formation, gradual growth, and ultimate fragmentation [20,22,24,25]. Despite this variability, the Bbs always comprise two essential elements: highly clustered mitochondria and irregular accumulations of electron dense granulo-fibrillar material, termed the nuage (Figure 1B,C) (molecular composition of nuage is discussed in [23,26,27,28]. Recent studies have indicated that the mitochondria accumulated within the Bb always exhibit distinctly higher inner membrane potential than these present in the ooplasm outside this organelle assemblage (Figure 1A) [19,29,30,31]. These observations lead to the idea that only highly functional mitochondria, containing the wild-type mtDNA are recruited to the Bb; here they proliferate and multiplicate [8,19,21]. This idea conforms well with classical EM autoradiographic studies showing that mtDNA present in the mitochondria of the Bb is intensely and synchronously replicated [32]. Further fate of the Bb mitochondria depends on the presence or absence of a specific oocyte region termed the germ or pole plasm [33]. If this oocyte region is absent, the Bb mitochondria (after dispersion of this organelle assemblage) populate the growing oocyte cytoplasm (the ooplasm) and are subsequently transmitted to the zygote that is to the next generation [31]. If the germ plasm is present the situation is more complicated: the Bb mitochondria are delivered not only to the ooplasm per se but also to this particular oocyte region; consequently they are inherited (also) by the grandchildren [20,21,33].

Participation of the Bb in the selection of female germline mitochondria has been additionally substantiated by computer aided 3D reconstruction of this organelle assemblage at the EM level [31,34]. This study has shown that in the oocytes of an apterygote insect *Thermobia domestica*, the Bb mitochondria form a hyperfused network. 3D reconstructions have shown additionally that in the neighborhood of the mitochondrial network, small apparently degenerating mitochondria are present (Figure 1B, encircled) [31]. During further stages of oogenesis, the Bb disperses and its mitochondria populate the entire ooplasm. The above results have led to the speculation that the Bb mitochondrial network, as similar networks in somatic cells, is implicated in the maintenance and inheritance of mtDNA. The following scenario of Bb participation in mitochondria selection and mtDNA maintenance [34] has been proposed:The early germline cells receive mitochondria in various functional states. They are not selected but partitioned to the individual germline cells stochastically as assumed in the bottleneck phenomenon.As oogenesis progresses, the mitochondria present in each germline cell gather next to the nucleus and form the Bb. Within the Bb, mitochondria multiply and fuse forming an extensive local network.Dysfunctional mitochondrial units (containing mutated mtDNA) are separated from the network and eliminated by mitophagy.After dispersal of the Bb selected (healthy) mitochondria populate the ooplasm.

The suggested scenario agrees well with current theoretical models of mitochondrial quality control, assuming that fusion, fission, and mitophagy are linked and cooperatively increase mitochondrial functionality (see [1,2,3,4,5] for the discussion). It is interesting to add here, that survey of both classical and recent literature not directly related to mitochondrial dynamics, brings additional evidence supporting the suggested scenario and therefore participation of the Bb in the selection of female germline mitochondria:Labelling with various antibodies and fluorescent probes showed that in the oocytes of a sea urchin, *Paracentrotus lividus* highly active mitochondria accumulate next to the germinal vesicle and here colocalize with vesicular acidic organelles (autophagosomes, autolysosomes) [35,36]. Thus, the arrangement of organelles in sea urchin oocytes mimics those in *Thermobia*.Autoradiographic studies revealed that in *Xenopus* Bb mtDNA is replicated synchronously “in neighboring mitochondria” [32] implying that in this species several smaller mitochondrial networks may coexist within single Bb.In murine oocytes the number of mitochondria remaining in physical contact has been estimated to increase from 21–58% during Bb formation [37]. Moreover, reevaluation of microphotographs published elsewhere (e.g., Figure 5B,F in [37]; 1E in [38]; 7B in [39]) suggests that the Bb mitochondria are also interconnected in mammals.It has been recently shown, using an allele-specific FISH approach, that mitochondrial fragmentation is responsible for the removal of mutated mtDNA in the female germline cells of *Drosophila* [40].

## 4. Silencing the Mitochondrial Activity—Low Energy Pathway

Although there are several experimental papers showing that oocyte mitochondria (or at least some of them) are functionally silenced [41,42,43,44], the low energy pathway is recently often brought into question. Theoretically two variants of this pathway are feasible: (1) the oocyte mitochondria remain inactive (or are inactivated) during the growth phase of oogenesis, and needed energy (in the form of ATP) is transferred to the oocyte from surrounding cells. Alternatively (2) the oocyte mitochondria are highly active during oogenesis, then eliminated and replaced by inactive (or less active) mitochondria from associated nurse cells. Obviously, the second variant can be employed only by the species characterized by meroistic ovaries (compare Section 2).

Inactive or “less active” subpopulations of mitochondria have been reported in the oocytes of various animal lineages including earthworms, nematodes, insects, and vertebrates [41,42,43,44,45,46]. According to the authors’ interpretation, the reduced (lowered) activity of oocyte mitochondria results in alleviated level of ROS and, therefore, protection of mtDNA against deleterious mutations. As it has been mentioned above, this mechanism is recently challenged [8]. The main reason of this criticism is still an unanswered question: how are the mitochondria with lowered activity chosen and transmitted to the zygote? In other words, what is the fate of the subpopulation of highly active mitochondria in this pathway?

## 5. Can High Energy and Low Energy Pathways Coexist in Related Animal Lineages?

To determine whether high and low energy pathways coexist in related animal lineages, we have performed EM studies of developing oocytes in representatives of two taxa of hemimetabolous insects (long-horned grasshoppers and earwigs) searching for manifestations of both pathways. We have chosen this insect subgroup because of two reasons. First, the ovaries of the Hemimetabola can be either panoistic or meroistic (see Section 2). This situation is beneficial as it allows correlation of pathways implicated in the transmission of mitochondria with ovarian categories. Second, the ooplasm (oocyte cytoplasm) of hemimetabolous insects is homogeneous and devoid of the germ (pole) plasm that participates (if present) in the transmission of mitochondria (compare Section 3). The lack of this ooplasm region implies that the mitochondria destined for transmission are not concentrated in a certain area but are scattered throughout the whole ooplasm that, in turn, greatly simplifies morphological analyses.

Long-horn grasshoppers (family Tettigonidae) are characterized by panoistic ovaries. Early oocytes of tettigonids comprise large Bbs in a juxtanuclear position (Figure 2A insert). The Bbs consist of two zones: perinuclear and cytoplasmic (Figure 2A and Figure 2A insert). In the perinuclear zone numerous polymorphic accumulations of nuage material, small bean-shaped mitochondria, and short endoplasmic reticulum (ER) cisternae are present (Figure 2A). In the cytoplasmic zone, mitochondria associate with nuage forming characteristic nuage/mitochondria complexes (Figure 2A). In each complex, the mitochondria are clustered around large, centrally located nuage accumulation (Figure 2). In addition, thin nuage filaments surround and link neighboring mitochondria (Figure 2B–D, arrowheads). Although the 3D reconstruction has not yet been completed, analysis of serial ultrathin sections suggests that the mitochondria associated with nuage elongate, bifurcate, and eventually form local micro-networks (Figure 2B–D). Interestingly, the mitochondria not directly associated with the nuage often show signs of degeneration (Figure 2C,D). As oogenesis progresses the nuage/mitochondria complexes are partitioned into progressively smaller entities that move towards oocyte cortex and consequently populate the whole ooplasm. Above morphological observations indicate that in tettigonids, as in other species mitochondria present in the Bb multiplicate and fuse. Our analyses suggest additionally that multiplication of the mitochondria within the Bb may be initiated by the contact with the nuage (unpublished observation).

Oocytes of earwigs (Dermaptera) are associated with single nurse cells, and therefore classified as meroistic [47,48]. Our analyses showed that in dermapterans, during accumulation of reserve materials, nearly all oocyte mitochondria are transported to the subcortical cytoplasm (Figure 3A). Here they colocalize with active Golgi complexes and abundant ER elements (Figure 3B). The result of this colocalization is quite surprising: all the mitochondria present in the subcortical region become surrounded by ER elements (Figure 3B,C). We suggest that in this way a process of mass mitophagy is initiated. This assumption is in line with two additional observations: (1) the mitochondria surrounded by ER lose cristae and (2) show signs of degeneration (Figure 3B,C). Our supposition is in agreement with current hypotheses assuming that mitophagy starts with the encasement of mitochondria between ER cisternae that provide a phagophore or a “cradle” for the formation of mitophagosomes (for further details see [49,50,51,52]). We hypothesize that in earwigs during the nurse cell dumping (compare Section 2), the ooplasm becomes repopulated by the less active mitochondria present in the cytoplasm of the nurse cell (Figure 3D). If our supposition is correct, the process described above represents the first example of the second variant of the low energy pathway. It should be underlined in this place that the proposed scenario is speculative and tentative; it must be tested and confirmed by additional supportive evidences.

## 6. Conclusions and Future Perspectives

On the basis of our EM studies, we suggest that the “high energy” and “low energy” pathways may operate even in closely related lineages of a given taxon. This, in turn, implies that these pathways are not evolutionary conserved and are rather related to morpho-functional characteristics of the ovaries as for instance: presence/absence of the nurse cells associated with the oocyte or presence of the Bb in the ooplasm.It is well known that in the ovaries of both vertebrates and invertebrates, a relatively large fraction of young germline cells is eliminated via apoptosis [53,54,55,56]. We believe, in this context that the selection of the mitochondria in oocytes is a two-step process: The first step operates at the cellular level and involves selection of mitochondria and mtDNA variants. In the second, all the germline cells that still (after the first step) contain large number of mutated mtDNA are eliminated via apoptosis. This idea is in line with our EM analyses showing that in apoptotic oocytes of *Thermobia*, the Bbs mitochondria reveal highly altered morphology (Figure 1C).The mechanisms responsible for mitochondrial inheritance obviously need further studies. There are at least two fundamental questions that should be answered: How replication of mtDNA (and consequent multiplication of mitochondria) located within the Bb is initiated? Is nuage material implicated in this initiation?Is recruitment of highly functional and healthy mitochondria to the Bb guaranteed by cytoskeleton (microtubules) as suggested by Milani [8] or it is an obvious consequence of a bottleneck phenomenon?

## Figures and Tables

**Figure 1 genes-11-00104-f001:**
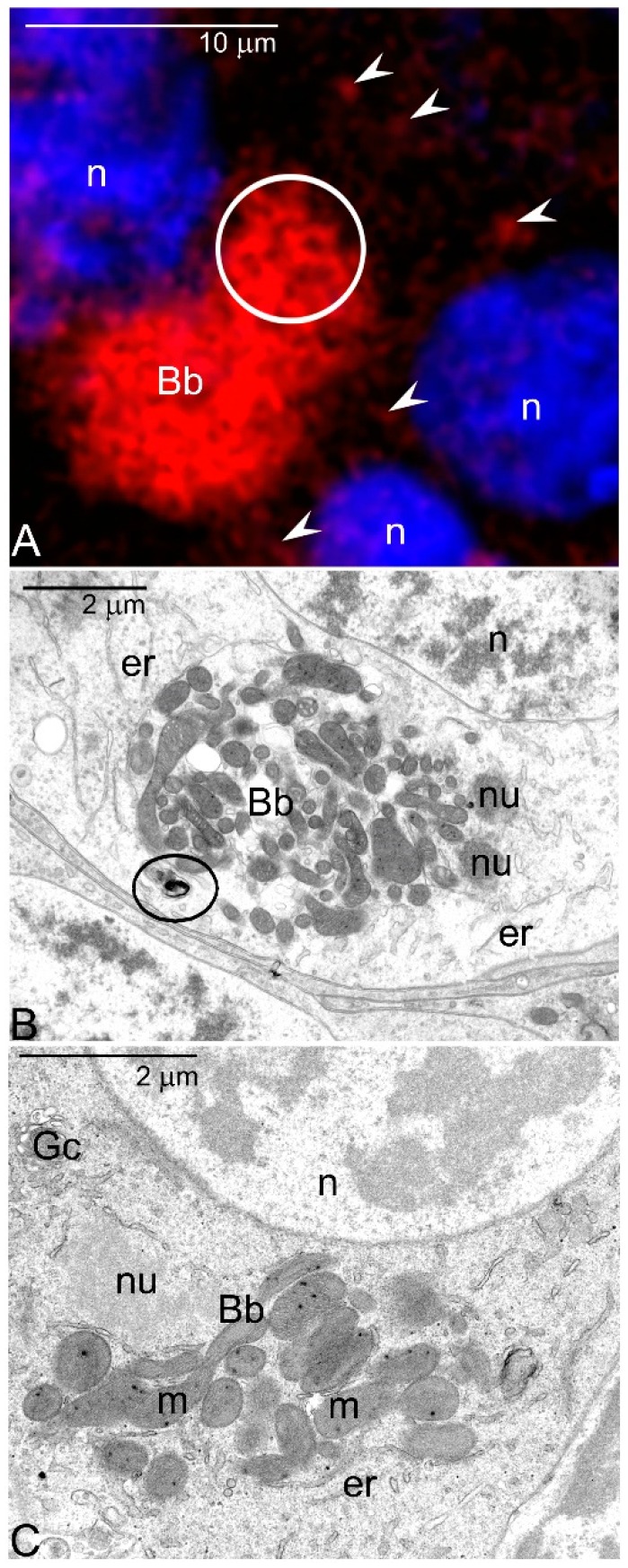
Balbiani body (Bb) of *Thermobia domestica*. (**A**) Meiotic oocyte incubated with MitoTracker^®^ Deep Red FM, counterstained with Hoechst 33342. Bb mitochondria are highly active, whereas those present outside this organelle assemblage exhibit lower activity (arrowheads). Note that that Bb mitochondria are elongated and ramified (encircled). Confocal microscope. (**B**) Medial section trough the Bb in developing oocyte. Note numerous mitochondria (m), accumulations of nuage material (nu), endoplasmic reticulum cisternae (er) and degenerating mitochondria outside the Bb (encircled). Transmission electron microscope (TEM). (**C**) Medial section through the Bb in apoptotic oocyte. Note nuage accumulation (nu), endoplasmic reticulum cisternae (er), Golgi complex (Gc), and swollen/enlarged mitochondria. TEM. Oocyte nucleus (n).

**Figure 2 genes-11-00104-f002:**
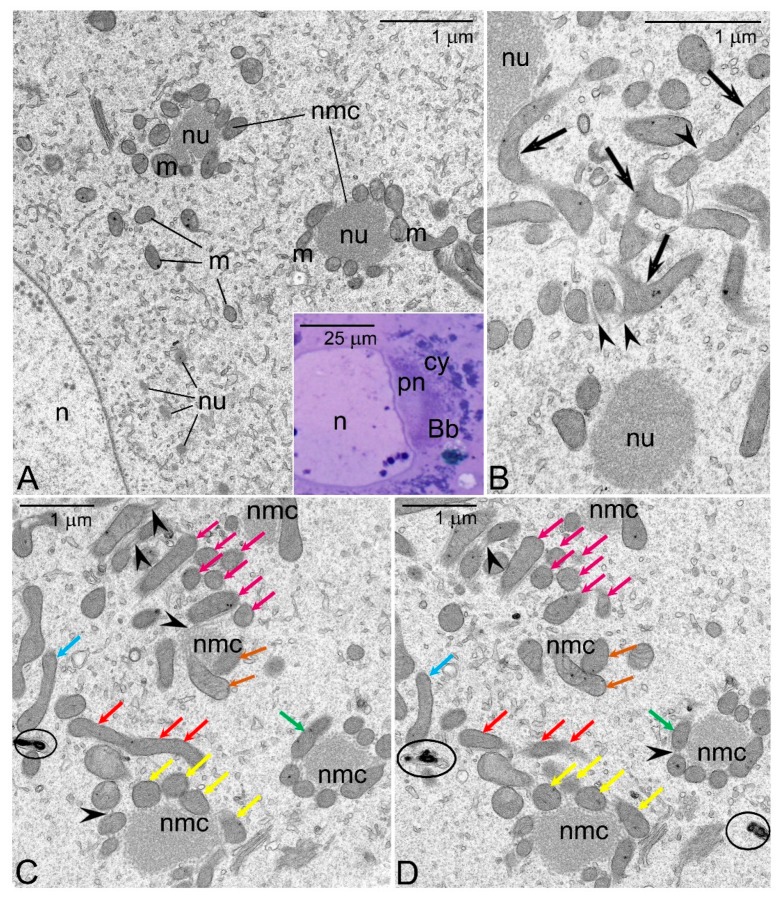
Balbiani body (Bb) of *Metrioptera brachyptera*. (**A**) and (**insert**) The Bb consists of two zones: perinuclear (pn) and cytoplasmic (cy). Note mitochondria (m), nuage accumulations (nu) and nuage/mitochondria complexes (nmc). Oocyte nucleus (n). TEM; (**insert**) semi-thin section stained with methylene blue. (**B**) Elongated and bifurcated mitochondria (arrows) in contact with nuage accumulations (nu) and nuage filaments (arrowheads). TEM. (**C**,**D**) Serial sections through four closely positioned nuage/mitochondria complexes (nmc). Small arrows (in various colors) indicate corresponding mitochondrial profiles, arrowheads point to nuage filaments, mitochondria displaying signs of degeneration are encircled. TEM.

**Figure 3 genes-11-00104-f003:**
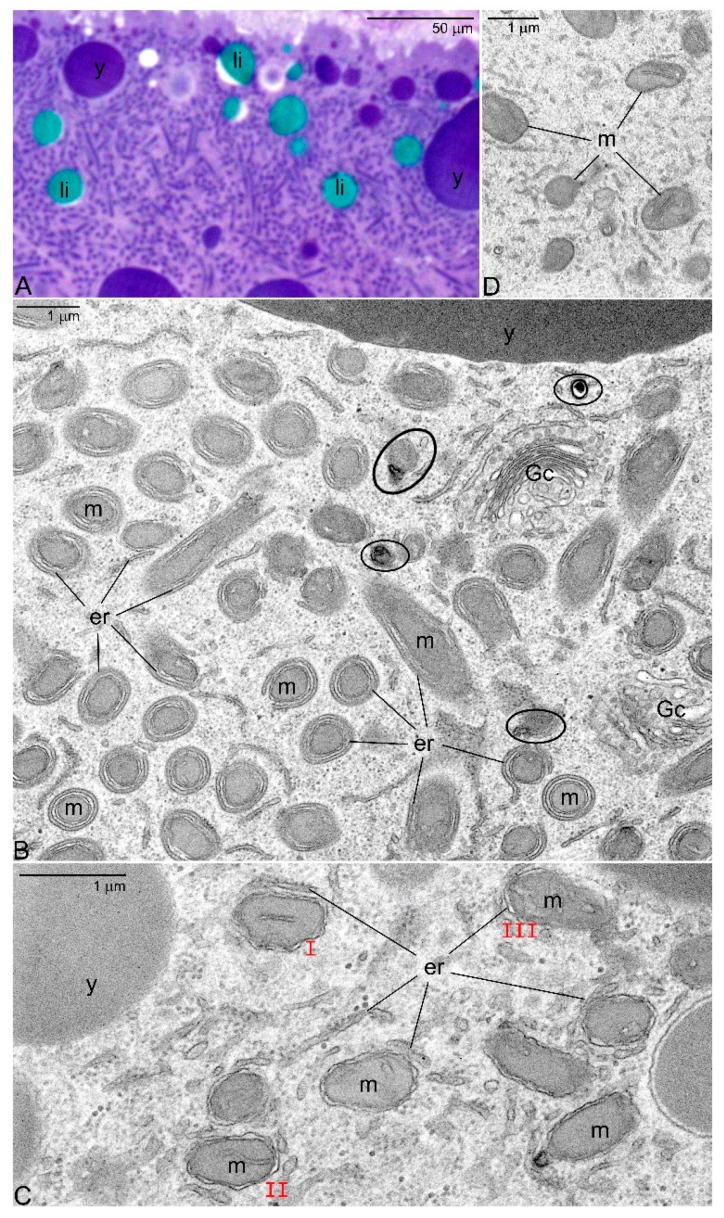
Elimination of mitochondria in vitellogenic oocytes of dermapteran, *Hamaxas nigrorufus*. (**A**) Semithin section through the subcortical ooplasm. Note numerous profiles of mitochondria. Methylene blue. (**B**) Ultrathin section of the same ooplasm region. Note that nearly all mitochondria (m) are tightly surrounded by endoplasmic reticulum cisternae and are devoid of cristae. Mitochondria displaying signs of degeneration are encircled. TEM. (**C**) The onset of mitophagy: mitochondria exhibit more or less classic morphology and still comprise cristae; the endoplasmic reticulum cisternae (er) start to associate with mitochondria surface; subsequent stages of this association (I–III). TEM. (**D**) Mitochondria in the nurse cell cytoplasm. TEM. Endoplasmic reticulum cisternae (er), Golgi complexes (Gc), lipid droplets (li), mitochondria (m), yolk granules (y).

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
