# Peer review of "Transmission of Functional, Wild-Type Mitochondria and the Fittest mtDNA to the Next Generation: Bottleneck Phenomenon, Balbiani Body, and Mitophagy"

_genes, 2020, doi:10.3390/genes11010104_

Round 1

Reviewer 1 Report

Uniparental, maternal in particular, inheritance of mitochondria is widely observed throughout the animal kingdom yet the mechanisms underlying uniparental transmission are not fully understood. This article evaluates evidence from the literature for distinct hypotheses regarding potential mechanisms of mitochondrial selection, including selection based on high or conversely low mitochondrial respiration and activity. In addition, the authors provide new comparative analyses of data from closely related animals to probe whether mechanisms of selection are evolutionarily conserved or not. Based on their review of the published literature and analyses the authors propose distinct mechanisms of mitochondrial selection exist. Moreover, they argue that the mode used in a given animal is likely more dependent on ovary type than relatedness, as two modes are detected in related animals 1) high energy and Balbiani body associated, and 2) low energy and somatic/nurse cell associated. Overall this review is interesting, well organized, well written and the figures are beautiful; however, there are a few unclear sentences and grammatical issues that need to be addressed. These are detailed below.

Points to address clarify:

The authors are encouraged to consider adding a model depicting the ovary type relationship and selection type. Recent work from Leiber et al should also be discussed. Leiber Nature. 2019 Jun;570(7761):380-384. doi: 10.1038/s41586-019-1213-4. Epub 2019 May 15. Maybe there is a mistake in the sentence on page 2 from line 65-67 because the nomenclature defined here is not intuitive and does not seem to match the previous sentence. In the sentence before "functionally silenced mitochondria" are described first so in this sentence appear to be the "high energy", but wouldn't these be the low energy mitochondria and the second population described in the sentence before "functionally highly active" be the high energy? Please clarify. The sentence on page 5 line 50 is unclear as written. Consider simplifying. For example: “After dispersal of the Bb, the selected (healthy) mitochondria populte the ooplasm.” The sentence of pg 5 lines 153-156 is a complicated and long sentence. Some of the language used is vague and consequently the key point is unclear. The sentence on pg 5 lines 161-163 is unclear as written. Consider simplifying. For example: “Autoradiographic studies of Xenopus Bb revealed the mtDNA replication is synchronous between neighboring mitochondria, implying that ....” (Also Xenopus should be italicized here) The sentence on pg 5 lines 164-167 is unclear as written. Consider simplifying. For example: "in murine oocytes the number of mitochondria remaining in physical contact has been estimated to increase from 21-58% during Bb formation (ref). Moreover, reevaluation of microphotographs published elsewhere...suggests that the Bb mitochondria are also interconnected in mammals.” The sentence on page 5 lines 169 -173 should be split into two sentences. Place a Theoretically two variants of this pathway are feasible: (1) the oocyte mitochondria (or at least some of them) remain inactive (or are inactivated) during the growth phase of oogenesis, and needed  energy (in the form of ATP) is transferred to the oocyte from surrounding (somatic or germline) cells. Alternatively (2) the oocyte mitochondria are highly active during oogenesis, then eliminated and  replaced by inactive (or less active) mitochondria from associated cells. Pg 6 line 186 Please be more specific than “to answer the title question”. For example: “ To determine if high and low energy pathways coexist in related animal lineages,….”

Minor editing:

Pg 1 line 24: “untouched” is probably not the right word here. Do the authors mean undamaged or not selected or both?

Pg 1 line 32: “lessions” should be “lesions”.

Pg1 line 40: “in form” should be “in the form”

Pg 2 line 48: “form” should be “from”:

Pg 2 line 60: “of oogonial” should be “of the oogonial”

Pg 2 line 63: “vivid debate which” should be “vivid debate about which”

Pg 2 line 64: Instead of “not affected mitochondria” consider “undamaged mitochondria”

Pg4 line 140 “of the Bb”  - no “the” is needed here.

Pg5 line 146 “muliplicate” should be “multiply”

Pg 5 line 158 “mitochondria accumulate” rather than “are accumulated”

Pg 6 line 190 Consider “This situation is beneficial as it allows…”

Pg 6 lines 202- 212 have several grammatical errors.

Pg 6 line 212 “in preparation” should be “unpublished observation”

Pg 8 line 235 “example” rather than “exemplification”

Pg 8 line 251 “implies” rather than “implicates”

Pg 9 lines 254-255 “a relatively large fraction of young germline cells are…”

Author Response

Rev. 1 (corrections marked green)

Unfortunately, it is too early to present a coherent model depicting ovary/selection type relationship. Our ideas, as underlined by Rev. 2, must be tested and confirmed by additional supportive evidences.  The paper of Leiber at al. (2019) on the role of mitochondrial fragmentation in selective removal of deleterious mutations in mtDNA was added and discussed (page 5, lines 167-169) according to Reviewer’s suggestion. All consecutive references have been renumbered. An obvious mistake in nomenclature of high and low energy pathway pointed out by the Reviewer has been corrected. All the contentious fragments have been re-written, all grammatical and typographical errors have been corrected accordingly suggestions and marked green in the corrected manuscript.

Reviewer 2 Report

In the manuscript titled “Two disparate pathways support transmission of functional, wild-type mitochondria and the fittest mtDNA to the next generation”, the authors reviewed the literatures that support  mtDNA selections that favor healthy or active mitochondria. They also formulated an argument for an alternative selection process, termed the low energy pathway that favor inactive or less active mitochondria. They also present data of TEM analyses on mitochondrial morphology and positioning in oocytes of several insect species to support their argument.  

On the first glance, I found the idea of a selection favoring low energy mitochondria unorthodox and baffling. There is no debate that organisms have to pass on healthy mitochondria that contain functionally competent mitochondrial genome to the next generation. The question is—how germ cells discern healthy mitochondria vs unfitted ones in a cell? Obviously cells do not have a mechanism to directly perceive which mtDNA molecule is wild type, which is a mutant (unless there is a mismatch or damage). Instead, they rely on the functional read out of mtDNA. In a heteroplasmic cell, mitochondria containing wild type genome will produce functionally-intact respiratory chain complexes and be polarized, whereas mutant mtDNA will lead to defective respiratory chain complexes and depolarized mitochondria. Currently, all proposed mechanisms including bottleneck inheritance, selective replication and Balbiani body mediated selection are under the premise that selections favor healthy or active mitochondria.

On the other hand, the authors’ hypothesis, particularly the supposition that mitochondria in the oocyte will be replaced by these in the nurse cell during earwigs oogenesis is quite intriguing. It reminds me of the sexual reproduction of some protists that have two nuclei: one macronucleus is actively transcribed to support all cellular activities; another micronucleus is transcriptionally silent but passed down to progeny cells, generating both micro- and macronucleus of the next generation. However, to propose a such provocative hypothesis as this, authors shall either present a logically cohesive argument, or provide supportive evidence. Regrettably, this manuscript fell short on both cases.

One premise of authors’ argument is that active mitochondria would produce excessive free radicals, which will damage the mtDNA. Therefore, silencing mitochondrial activity would be helpful to maintain the integrity of mtDNA. They referenced previously studies showing that there are two distinct mitochondrial populations in mammalian oocytes and early embryos. One group of mitochondria are active, while another group of mitochondria are inactive. But there is no evidence these low energy mitochondria are preferentially utilized to populate mitochondria in oocytes, or furnish germ plasm. On the contrary, the model of Balbiani body argues that healthy or active mitochondria are selected to populate oocytes. Perhaps, authors should ask why mitochondria have distinct activities within a single oocyte or embryo. Given that all mitochondria would receive the same set of nuclear-encoded mitochondrial proteins in a cell, the functional discrepancy would most likely reflect the difference on mtDNA. Wild type mtDNA would lead to active mitochondria, while mtDNA mutations may lead inactive, low-energy mitochondria. Therefore, it does not make any sense to select for low-energy mitochondria.

I do find the TEM images showing that almost all mitochondria are enveloped by ER membrane in earwigs ovaries (Figure 3) fascinating. I am not an expert, and do not know much about the development and physiology of this organism. The ER-mitochondria contact may not simply indicate mitophagy. It is known that mitochondria-ER contact promotes mtDNA replication. Are mitochondria in the oocyte truly eliminated through autophagy and replaced by these in the nurse cell? Do mitochondria in the nurse cell and in the oocyte have distinct activities… There are many unanswered questions that I hope the authors would look into before making any big statements.

Author Response

Rev. 2 (corrections marked yellow)

We agree with the reviewer that the idea of a selection favoring low energy mitochondria is not appropriately supported. Therefore all fragments dealing with the “low energy pathway” have been rewritten and corrected. As to our speculative (“provocative”) supposition that oocyte mitochondria can be replaced by nurse cell mitochondria - we agree that it should (or must) be additionally tested and confirmed. Corresponding paragraphs of the text have been rewritten and corrected accordingly. However, we have to mention that the transport of mitochondria from the nurse cells to the oocyte is a well documented phenomenon. It has been recently shown that this phenomenon is characteristic not only for numerous invertebrates but also for certain mammals (i.e. mice). The information about this transport is included in section 2 “Categories (types) of animal ovaries”. This section has been slightly modified to clarify the situation. We realize that ER-mitochondria contact sites might be implicated in various physiological processes, however, revealed in our analyses apparently altered morphology of mitochondria (disappearance of cristae and morphological signs of degeneration), suggest that the process of mitophagy has been initiated. In consequence of numerous modifications of the text, we have decided to change the title of the manuscript into more neutral one.

Round 2

Reviewer 2 Report

The authors have satisfactorily addressed most of the issues I raised earlier. One minor point, for a comprehensive review on mtDNA inheritance, the authors shall include another model of selection, the selective enrichment of healthy mitochondria through replication competition that has been demonstrated in Drosophila (Zhang et al., 2019 Mol. Cell).